# VARIANCE-BASED GRADIENT COMPRESSION FOR EFFICIENT DISTRIBUTED DEEP LEARNING

## ABSTRACT

Due to the substantial computational cost, training state-of-the-art deep neural networks for large-scale datasets often requires distributed training using multiple computation workers. However, by nature, workers need to frequently communicate gradients, causing severe bottlenecks, especially on lower bandwidth connections. A few methods have been proposed to compress gradient for efficient communication, but they either suffer a low compression ratio or significantly harm the resulting model accuracy, particularly when applied to convolutional neural networks. To address these issues, we propose a method to reduce the communication overhead of distributed deep learning. Our key observation is that gradient updates can be delayed until an unambiguous (high amplitude, low variance) gradient has been calculated. We also present an efficient algorithm to compute the variance and prove that it can be obtained with negligible additional cost. We experimentally show that our method can achieve very high compression ratio while maintaining the result model accuracy. We also analyze the efficiency using computation and communication cost models and provide the evidence that this method enables distributed deep learning for many scenarios with commodity environments.

## 1 INTRODUCTION

Deep neural networks are attracting attention because of their outstanding prediction power in many application fields such as image recognition, natural language processing, and speech recognition. In addition, software frameworks are publicly available, making it easier to apply deep learning. However, their crucial drawback is the substantial computational cost on training. For example, it takes over a week to train ResNet-50 on the ImageNet dataset if using a single GPU. Such long training time limits the number of trials possible when creating models.

Therefore, we must conduct distributed training using multiple computation workers (e.g., multiple GPUs in different nodes). However, by nature, workers need to frequently communicate gradients, which yields a severe bottleneck for scalability, especially when using lower bandwidth connections. For example, when using 1000BASE-T Ethernet, communication takes at least ten times longer than forward and backward computation for ResNet-50, making multiple nodes impractical. High performance interconnections such as InfiniBand and Omni-Path are an order of magnitude more expensive than commodity interconnections, which limits research and development of deep learning using large-scale datasets to a small number of researchers.

Although several methods have been proposed to compress gradient for efficient communication, they either suffer a low compression ratio or significantly harm the resulting model accuracy, particularly when applied to convolutional neural networks. There are mainly two lines of research: quantization and sparsification. Quantization-based methods include *1-bit SGD* (Seide et al., 2014) and *TernGrad* (Wen et al., 2017). Though they achieve small loss of accuracy by using at least one bit for each parameter, the compression ratio is limited. Sparsification-based methods include Strom (2015) and *QSGD* (Alistarh et al., 2017). While they can achieve high compression ratio, as we will see in our experiments, they harm the resulting model accuracy or suffer a low compression ratio, particularly when applied to convolutional neural networks.

To address these issues, we propose a new gradient compression algorithm to reduce the communication overhead of distributed deep learning. The proposed method belongs to the sparsification

approaches. Our key observation is that the variance of the gradient for each parameter point over iterations is a useful signal for compression. As almost all previous approaches of both sparsification and quantization only look at the magnitude of gradient, we believe that we are opening a new door for this field. In addition, we also show that our method can be combined with previous compression methods to further boost performance. We also present an efficient algorithm to compute the variance and prove that it can be obtained with negligible additional cost.

We experimentally demonstrate that our method can achieve a high compression ratio while maintaining result model accuracy. We also analyze the efficiency using computation and communication cost models and provide evidence that our method enables distributed deep learning for many scenarios with commodity environments.

**Organization.** The remainder of this paper is organized as follows: Section 2 provides the definitions and notations used in this paper. Section 3 reviews related work in this field. Section 4 presents the proposed method. Section 5 analyzes performance. Section 6 shows our experimental results, and we conclude in Section 7.

## 2 PRELIMINARIES

In this section, we describe an overview of distributed deep learning and parameter updates with compressed gradients.

### 2.1 CHALLENGES IN DATA PARALLEL STOCHASTIC GRADIENT DESCENT

In data parallel distributed Stochastic Gradient Descent (SGD), all workers have identical copies of the same model and calculate gradients using different subsets of training data. Gradients are shared across all workers, and each worker updates its local model using the shared gradients.

There are two well-known approaches to communication of gradients: synchronous and asynchronous. Even though our method can be applied to both of them, we focus on the synchronous approach in this paper. Each worker computes gradients and shares them with other workers using a synchronized group communication routine in every single training iteration, typically using a communication routine known as allreduce.

The challenge is that the communication is possibly a severe bottleneck in a training process. Gradients typically consist of tens of millions of floating point values so the total size of exchanged data can be large. For example, the model size of ResNet-50 (He et al. (2016)) is over 110 MB, and the size of gradients becomes large accordingly. Thus, the communication time has a significant effect on the total training time in environments equipped with a commodity interconnect hardware, such as 1Gb Ethernet. Also, in the synchronous approach, workers have to wait for completion of communication and their computing resources including GPUs are idle, which is a significant performance loss.

### 2.2 PROBLEM FORMULATION

In basic procedures of SGD, model parameters are updated as

$$x_{t+1} = x_t - \gamma \nabla f_t(x_t),$$

where $x_t$ and $\nabla f_t(x_t)$ are model parameters and calculated gradients in time step $t$, respectively. $f_t$ is a loss function and it differs between samples used in a mini-batch. $\gamma$ is a step size.

To reduce an amount of data to be exchanged over a network, either quantization or sparsification or both are used as explained in Sec. 3.

## 3 RELATED WORK

There are two main approaches to gradient compression: quantization-based approaches and sparsification-based approaches. Quantization-based approaches reduce communication cost by expressing each gradient with fewer bits. If a baseline uses 32-bit floating points in communication,

then it can reduce the amount of communication by up to 32 times. Seide et al. (2014) showed that neural networks can be trained using only one sign bit per parameter. There are two key techniques in their algorithm. First, they use different threshold to encode and decode gradient elements for each column of weight matrix. Second, quantization errors are added to the gradients calculated in the next step. Its effectiveness has been experimentally verified through speech models. Wen et al. (2017) proposed TernGrad to encode gradients with 2 bits per parameter. The algorithm is characterized by its theoretically-guaranteed convergence and reported that it can successfully train GoogLeNet (Szegedy et al., 2015) on ImageNet with an average loss of accuracy of less than 2%.

As a second approach, sparsification-based approaches reduce communication cost by sending only a small fraction of gradients. Even though they require sending not only the values of gradients but also parameters' indexes, their strong sparsification reduces transmission requirements significantly. Strom (2015) proposed sending only gradients whose absolute values are greater than a user-defined threshold. The algorithm sends only sign bits and encoded indexes of parameters. Gradients are decoded up to the threshold and quantization errors are added to the gradients calculated in the next step as 1-bit stochastic gradients. Its effectiveness has also been experimentally verified on speech applications. Dryden et al. (2016) extended Strom's method. They proposed to use an adaptive threshold instead of using a user-defined threshold. They also introduced repeated sparsification of gradients in order to combine the algorithm with an efficient communication algorithm. Alistarh et al. (2017) proposed QSGD. QSGD stochastically rounds gradients to linearly quantized values, which are calculated in the algorithm. Their work enjoys strong theoretical properties in convex optimizations. Furthermore, they can control the trade-off between accuracy and compression. On the other hand, Strom's method does not work with small or large thresholds.

## 4 PROPOSED METHODS

In this section, we describe the proposed method. An efficient implementation and combination with other compression methods are also explained.

Our work belongs to the sparsification-based approaches. In this section, we explicitly denote a gradient vector ($\nabla f(x)$) and a gradient element ($\nabla_i f(x)$) for clarity. Previous works in this direction have focused on gradient elements with small magnitudes, and they rounded them to zero to sparsify. Our work diverges at this point. We propose using approximated variances of gradient elements instead of magnitudes. Our method do not transmit ambiguous elements until additional data reduce their ambiguity and significantly reduces communication while maintaining accuracy. This method enables shifting the balance between accuracy and compression, as necessary. Furthermore, we can combine our work with sparsity-promoting quantization like QSGD and Strom's method. We show the way of combination with the Strom's method later.

### 4.1 KEY CONCEPTS

The key idea of our method is delaying sending ambiguously estimated gradient elements. We consider a gradient element to be ambiguous when its amplitude is small compared to its variance over the data points. We extend the standard updating method to the following:

$$x_{t+1} - \gamma r_{t+1} = x_t - \gamma(\nabla f_t(x_t) + r_t).$$

This extension follows Seide et al. (2014) and Strom (2015).

In previous works, the approximation errors are accumulated in $r_t$ and used in future updates. In each step, parameters are updated only with approximated gradient elements represented by less number of bits. In our work, we interpret $r_t$ as a delayed update, not approximation errors.

We send the gradient element corresponding to the $i$-th parameter only when it satisfies the following criterion,

$$\frac{\alpha'}{|B|} V_B[\nabla_i f_z(x)] < (\nabla_i f_B(x))^2, \tag{1}$$

where $|B|$ is a size of the mini-batch and $\alpha'$ is a hyper parameter representing required estimation accuracy. $z$ is a each sample and $B$ is a mini-batch, respectively and $f_z$ and $f_B$ are corresponding

loss functions. $V_B[\nabla_i f_z(x)]$ is the sample variance of the gradient element corresponding to the $i$-th parameter over a mini-batch $B$.

If we do not send some gradient elements, we add them to the next batch and recalculate (1) with increased batch size. For example, if we postpone sending a gradient element nine times consecutively, the criterion (1) is calculated as if ten times larger batch than usual mini-batch in the next step. Note that even though the criterion (1) is calculated as if we used a larger batch, what is used for an update is not the mean of the mini-batches across steps but the sum of them.

Following lemma supports our formulation.

**Lemma 4.1.** *(De et al., 2017) A sufficient condition that a vector $-g$ is a descent direction is*

$$\|g - \nabla f(x)\|_2^2 < \|g\|_2^2. \tag{2}$$

We are interested in the case of $g = \nabla f_B(x)$, the gradient vector of the loss function over $B$. By the weak law of large numbers, when $|B| > 1$, the left hand side of Eq. 2 with $g = \nabla f_B(x)$ can be estimated as follows:

$$E[\|\nabla f_B(x) - \nabla f(x)\|_2^2] \sim \frac{1}{|B|} V_B[\nabla f_z(x)].$$

Thus our formulation with $\alpha' \geq 1$ corresponds to an elementwise estimation of the sufficient condition (2) that a gradient vector decreases the loss function. Gradient elements become more likely to be sent as sample size increases. However, if once gradient elements are estimated with too high variances, it takes too long for the elements to be sent. Thus, we decay variance at every step. Details are described in subsection 4.4. In the combination with optimization methods like Momentum SGD, gradient elements not sent are assumed to be equal to zero.

## 4.2 QUANTIZATION AND PARAMETER ENCODING

To allow for comparison with other compression methods, we propose a basic quantization process. In this section, we refer to a gradient as an accumulated gradient. After deciding which gradient elements to send, each worker sends pairs of a value of a gradient element and its parameter index as Strom (2015) and Alistarh et al. (2017). We quantize each element to 4-bit so that we can represent each pair in 32-bit as per Strom (2015). The 4-bit consists of one sign bit and three exponent bits.

Our quantization except for the sign bit is as follows. For a weight matrix $W_k$ (or a weight tensor in CNN), there is a group of gradient elements corresponding to the matrix. Let $M_k$ be the maximum absolute value in the group. First, for each element $g_i$ in the $k$-th group, if $|g_i|$ is larger than $2^{\lfloor \log_2 M_k \rfloor}$ truncate it to $2^{\lfloor \log_2 M_k \rfloor}$, otherwise, round to the closer value of $2^{\lfloor \log_2 |g_i| \rfloor}$ or $2^{\lceil \log_2 |g_i| \rceil}$. Let $g_i'$ be the preprocessed gradient element. Next, calculate a integer $d_i := \lfloor \log_2 M_k \rfloor - \log_2 g_i'$. If $d_i > 7$ then we do not send the value, otherwise, encode the integer from 0 to 7 using 3 bits. $\lfloor \log_2 M_k \rfloor$ is also sent for every weight matrix. An efficient implementation is presented in subsection 4.4. We do not adopt stochastic rounding like Alistarh et al. (2017) nor accumulate rounding error $g_i - g_i'$ for the next batch because this simple rounding does not harm accuracy empirically. Appendix B has a running example of this quantization.

Because the variance-based sparsification method described in subsection 4.1 is orthogonal to the quantization shown above, we can reduce communication cost further using sparsity promoting quantization methods such as QSGD instead. However, we used the quantization to show that enough level of sparsity is gained solely by our variance-based sparsification because the quantization rounds only a small fraction of gradient elements to zero. We show how to combine our method with a method in Strom (2015) later in this paper because the way of the combination is less obvious. We use a naive encoding for parameter indexes because the rest 28-bits are enough. We can further reduce the number of bits by compressing parameter indexes (Strom, 2015; Alistarh et al., 2017).

## 4.3 COMMUNICATION BETWEEN WORKERS

In distributed deep learning, the most important operation is to take the global mean of the gradient elements calculated in each worker. The operation is referred to as "allreduce." It consists of three steps: (1) collects all local arrays in each worker, (2) reduce them using a given arithmetic operator,

which is summation in this case, and (3) broadcast the result back to all workers so that all workers obtain the identical copies of the array.

Conventional data parallel deep learning applications can enjoy the benefit of highly optimized allreduce implementations thanks to the fact that only the sum of the values has to be kept during the communication. However, after applying the proposed method to the local gradient elements, they are converted to a sparse data structure so the allreduce operation can no longer apply.

Dryden et al. (2016) and Aji & Heafield (2017) proposed sparsifying gradient elements multiple times to utilize a kind of allreduce for sparsification-based compressions. However, the accuracy is possibly degraded when the elements are highly compressed through repetitive compressions. Instead, we adopt allgatherv for communication, where each worker just sends the calculated elements to other workers. We avoid to encode and decode elements multiple times by allgatherv. In allgatherv communication cost hardly increase from a kind of allreduce because index overlap, which is needed for summation, rarely occur if the compression ratio is sufficiently higher than the number of workers. Thanks to the high compression ratio possible with this algorithm and its combination with other compression methods, even large numbers of workers can be supported. Some optimization methods, such as ADAM (Ba & Kingma, 2015), require parameter updates and postprocessing. They are calculated locally after the communication.

## 4.4 Efficient implementation

We first describe the efficient computation of the criterion (1) and second how to quantize gradient elements without additional floating points operations. We can efficiently compare squared mean and variance in the criterion (1) by just comparing squared mean of gradient elements and sum of squared gradient elements. That is,

$$\left(\sum_{z \in B} \frac{1}{|B|} \nabla_i f_z(x)\right)^2 > \alpha \sum_{z \in B} \left(\frac{1}{|B|} \nabla_i f_z(x)\right)^2 \tag{3}$$

achieves our goal. Thus, we have to maintain only the sum of gradient elements and sum of squared gradient elements. Details are described in Appendix A. Decay of variance, described in subsection 4.1, is accomplished by multiplying hyperparameter $\zeta(< 1)$ to the sum of squared gradient elements at every step. Alpha in Eq. 3 controls how much unambiguity the algorithm require. The algorithm compress more aggressively with larger alpha. A range from one to two is good for alpha from its derivation. Fig. 1 shows the final algorithm.

The quantization of parameters described in subsection 4.2 can also be efficiently implemented with the standard binary floating point representation using only binary operations and integer arithmetic as follows. We can calculate $2^{\lfloor \log_2 x \rfloor}$ by truncating the mantissa. We can also round values by adding one to the most significant bit of mantissa as if $x$ is an unsigned integer and then masking mantissa to 0.

## 4.5 Hybrid algorithm

We describe the way to combine our method with Strom's method. It becomes problematic how to modify variance when only parts of gradient elements are exchanged. We solve the issue simply by modifying $a^2$ to $(a-b)^2$. Let S be the value sent in a step (i.e. threshold, -threshold or 0). We correct squared gradient elements $\sum(\nabla_i f)^2$ to $\sum(\nabla_i f)^2 - 2S \sum(\nabla_i f) + S^2$. Fig. 2 shows the algorithm. We show the effectiveness of this combined algorithm by experiments in Sec. 6. Combinations with other works like QSGD and TernGrad are rather straightforward and we do not explore further in this paper.

## 5 Performance analysis

Because common deep learning libraries do not currently support access to gradients of each sample, it is difficult to contrast practical performance of an efficient implementation in the commonly used software environment. In light of this, we estimate speedup of each iteration by gradient compression with a performance model of communication and computation. The total speed up of the whole training process is just the summation of each iteration because we adopt a synchronized approach.

**Algorithm 1:** Basic

**hyperparam:** $B$ = batch size, $\zeta, \alpha$
**foreach** *parameters*:
 $r_i = 0$;
 $v_i = 0$;
**while** *not converged*:
 CalcGrad();
 **foreach** *parameters*:
  $r_i \mathrel{+}= \sum \frac{\nabla_i f_z}{B}$;
  $v_i \mathrel{+}= \sum (\frac{\nabla_i f_z}{B})^2$;
  **if** $r_i^2 > \alpha v_i$:
   Encode($r_i$);
   $r_i = 0$;
   $v_i = 0$;
  **else:**
   $v_i \mathrel{*}= \zeta$;
 CommunicateAndUpdate();

Figure 1: Basic algorithm of our variance-based compression. $\zeta$ and $\alpha$ are hyperparameters. Recommended value for $\zeta$ is 0.999. $\alpha$ controls compression and accuracy. $\nabla_i f_z$ denotes a gradient element of parameter $i$ for each sample $z$ in mini-batch. CalcGrad() is backward and forward computaion. Encode() includes quantization and encoding of indexes. CommunicateAndUpdate() requires sharing gradient elements and decode, then update parameters.

**Algorithm 2:** Hybrid

**hyperparam:** $B$ = batch size, $\zeta, \alpha, \tau$
**foreach** *parameters*:
 $r_i = 0$;
 $v_i = 0$;
**while** *not converged*:
 CalcGrad();
 **foreach** *parameters*:
  $r_i \mathrel{+}= \sum \frac{\nabla_i f_z}{B}$;
  $v_i \mathrel{+}= \sum (\frac{\nabla_i f_z}{B})^2$;
  **if** $|r_i| > \tau$ *and* $r_i^2 > \alpha v_i$:
   Encode(Sign($r_i$));
   $r_i \mathrel{-}= \text{Sign}(r_i)\,\tau$;
   $v_i =$
    $\max(v_i - 2|r_i|\tau + \tau^2, 0)$;
  $v_i \mathrel{*}= \zeta$;
 CommunicateAndUpdate();

Figure 2: Hybrid algorithm of our variance-based compression and Strom's method. $\tau$ is a user-defined threshold required in Strom's method. Other parameters and notations are the same with Fig. 1.

In the communication part, the pairs of the quantized values and the parameter indexes in each node are broadcast to all nodes. The $i$-th node's input data size $n_i (i = 1, ..., p)$ may be different among nodes, where $p$ denotes the number of nodes. An MPI function called allgatherv realizes such an operation. Because recent successful image recognition neural network models like VGG (Simonyan & Zisserman, 2015) or ResNet (He et al., 2016) have a large number of parameters, the latency term in communication cost can be ignored even if we achieve very high compression ratio such as $c > 1,000$. In such cases, a type of collective communication algorithms called the ring algorithm is efficient (Thakur et al., 2005). Its bandwidth term is relatively small even though its latency term is proportional to $p$. Although the naive ring allgatherv algorithm costs unacceptable $O(\max_i n_i \cdot p)$ time, Träff et al. (2008) proposed a method to mitigate it by dividing large input data, which is called pipelined ring algorithm. For example, an allgatherv implementation in MVAPICH adopts a pipelined ring algorithm for large input data.

The calculation of variance of gradients dominates in the additional cost for the computation part of the proposed method. The leading term of the number of multiply-add operations in it is $2N|B|$, where $N$ and $|B|$ are the number of parameters and the local batch size, respectively. Other terms such as determination of sent indexes and application of decay are at most $O(N)$. Therefore hereafter we ignore the additional cost for the computation part and concentrate to the communication part.

We discuss the relationship between the compression ratio and the speedup of the communication part. As stated above, we ignore the latency term. The baseline is ring allreduce for uncompressed gradients. Its elapsed time is $T_r = 2(p - 1)Ns\beta/p$, where $s$ and $\beta$ are the bit size of each parameter and the transfer time per bit, respectively. On the other hand, elapsed time of pipelined ring allgatherv is $T_v = \sum \lceil ((n_i/m) - 1)m\beta \rceil$, where $m$ is the block size of pipelining. Defining $c$ as the averaged compression ratio including change of the number of bits per parameter, $T_v$ is evaluated as

$T_v \leq (\sum n_i + (p-1)m)\beta = (Nsp/c + (p-1)m)\beta$. If we set $m$ small enough, relative speedup is $T_r/T_v \geq 2(p-1)c/p^2$. Therefore we expect linear speedup in $c > p/2$ range.

# 6 EXPERIMENTS

In this section, we experimentally evaluate the proposed method. Specifically, we demonstrate that our method can significantly reduce the communication cost while maintaining test accuracy. We also show that it can reduce communication cost further when combined with other sparsification methods, and even improves test accuracy in some settings.

We used CIFAR-10 (Krizhevsky, 2009) and ImageNet (Russakovsky et al., 2015), the two most popular benchmark datasets of image classification. We fixed the hyperparameter $\zeta$ in Fig. 1 and Fig. 2 to 0.999 in all experiments. We evaluated gradient compression algorithms from the following two viewpoints: accuracy and compression ratio. The accuracy is defined as the test accuracy at the last epoch, and the compression ratio is defined as the number of the total parameters of networks divided by the average number of parameters sent. We do not consider the size of other non-essential information required for the communication, because they are negligible. In addition, we can ignore the number of bits to express each gradient because we assume that both a gradient and a parameter index are enclosed in a 32 bit word as Strom (2015) in all algorithms. Please note that, as all methods use allgatherv for communication, communication cost increases in proportion to the number of workers. Thus, high compression ratio is required to achieve sufficient speed up when using tens or hundreds of workers. We have visualization of results in Appendix C.

## 6.1 CIFAR-10

For experiments on CIFAR-10, we used a convolutional neural network similar to VGG (Simonyan & Zisserman, 2015). The details of the network architecture are described in Appendix D. We trained the network for 300 epochs with weight decay of 0.0005. A total number of workers was 8 and batch size was 64 for each worker. We applied no data augmentation to training images and center-cropped both training and test images into 32x32. We used two different optimization methods: Adam (Ba & Kingma, 2015) and momentum SGD (Sutskever et al., 2013). For Adam, we used Adam's default parameter described in Ba & Kingma (2015). For momentum SGD, we set the initial learning rate to $0.05 \times 8$ and halved it at every 25 epochs. We used two's complement in implementation of QSGD and "bit" represents the number of bits used to represent each element of gradients. "d" represents a bucket size. For each configuration, we report the median of the accuracy from five independent runs. Compression ratios are calculated based on the execution that achieved the reported accuracy.

Table 1 summarizes the results. Our method successfully trained the network with slight accuracy gain for the Adam setting and 2 to 3 % of accuracy degradation for the Momentum SGD setting. Compression ratios were also sufficiently high, and our method reduced communication cost beyond quantization-based approaches described in section 3. The hybrid algorithm's compression ratio is several orders higher than existing compression methods with a low reduction in accuracy. This indicates the algorithm can make computation with a large number of nodes feasible on commodity level infrastructure that would have previously required high-end interconnections. Even though QSGD achieved higher accuracy than our method, its compression power is limited and our algorithm can reduce communication cost more aggressively. On the other hand, Strom's method caused significant accuracy degradation. Counter-intuitively, the hybrid algorithm improved its accuracy, in addition to the further reduction of communication.

Our hypothesis for this phenomena is as follows. In Strom's algorithm, when a large positive gradient appears, it has no choice but send positive values for consequent steps even if calculated gradients in following mini-batches have negative values. On the other hand, in the hybrid algorithm, if following gradients have a different sign with a residual, the residual is not likely to be sent. We assume that this effect helped the training procedure and led to better accuracy. We also would like to mention the difficulty of hyperparameter tuning in Strom's method. As Table 1 shows, using lower threshold does not necessarily always lead to higher accuracy. This is because the hyperparameter controls both its sparsification and quantization. Thus, users do not know whether to use a larger or smaller value as a threshold to maintain accuracy. We note that we observed unstable behaviors

Table 1: Training of a VGG-like network on CIFAR-10. $\tau$ denotes the threshold in Strom's method. $\alpha$ is the hyperparameter of our method described in the criterion (3). The number of bits of QSGD refers the number of bits to express gradients except for the sign bits. For each configuration, the median of the accuracy from five independent runs is reported. The compression column lists the compression ratio defined at the beginning of Sec. 6.

| | Adam | | Momentum SGD | |
|---|---|---|---|---|
| **Method** | **Accuracy** | **Compression** | **Accuracy** | **Compression** |
| no compression | 88.1 | 1 | 91.7 | 1 |
| Strom, $\tau = 0.001$ | 62.8 | 88.5 | 84.8 | 6.5 |
| Strom, $\tau = 0.01$ | 85.0 | 230.1 | 10.6 | 990.7 |
| Strom, $\tau = 0.1$ | 88.0 | 6,942.8 | 71.6 | 8,485.0 |
| our method, $\alpha = 1$ | 88.9 | 120.7 | 90.3 | 52.4 |
| our method, $\alpha = 1.5$ | 88.9 | 453.3 | 89.6 | 169.2 |
| our method, $\alpha = 2.0$ | 88.9 | 913.4 | 88.4 | 383.6 |
| hybrid, $\tau = 0.01, \alpha = 2.0$ | 85.0 | 1,942.2 | 87.6 | 983.9 |
| hybrid, $\tau = 0.1, \alpha = 2.0$ | 88.2 | 12,822.4 | 87.1 | 12,396.8 |
| QSGD (2bit, $d = 128$) | 88.8 | 12.3 | 90.8 | 6.6 |
| QSGD (3bit, $d = 512$) | 87.4 | 14.4 | 91.4 | 7.0 |
| QSGD (4bit, $d = 512$) | 88.2 | 11.0 | 91.7 | 4.0 |

Table 2: Training ResNet50 on ImageNet. $\tau$ denotes a threshold in Strom's method. $\alpha$ is the hyperparameter of our method described in the criterion (3). Accuracy is the test accuracy at the last epoch. Compression refers compression ratio defined in the beginning of Sec. 6.

| | Adam | | Momentum SGD | |
|---|---|---|---|---|
| **Method** | **Accuracy** | **Compression** | **Accuracy** | **Compression** |
| no compression | 56.2 | 1 | 76.0 | 1 |
| Strom, $\tau = 0.001$ | 28.6 | 38.6 | 75.2 | 2.1 |
| Strom, $\tau = 0.01$ | 50.0 | 156.2 | 75.5 | 35.2 |
| Strom, $\tau = 0.1$ | 48.1 | 6,969.0 | 75.5 | 2,002.2 |
| our method, $\alpha = 1$ | 55.3 | 1,542.8 | 74.7 | 103.8 |
| our method, $\alpha = 1.5$ | 57.4 | 2,953.1 | 75.5 | 400.7 |
| our method, $\alpha = 2.0$ | 57.8 | 5,173.8 | 75.1 | 990.7 |
| hybrid, $\tau = 0.01, \alpha = 2.0$ | 52.2 | 2,374.2 | 75.0 | 470.9 |
| hybrid, $\tau = 0.1, \alpha = 2.0$ | 43.1 | 28,954.2 | 75.1 | 4,345.0 |

with other thresholds around 0.01. On the other hand, our algorithm are free from such problem. Moreover, when we know good threshold for Strom's algorithm, we can just combine it with ours to get further compression.

## 6.2 IMAGENET

As larger scale experiments, we trained ResNet-50 (He et al., 2016) on ImageNet. We followed training procedure of Goyal et al. (2017) including optimizer, hyperparameters and data augmentation. We also evaluated algorithms with replacing MomentumSGD and its learning rate scheduling to Adam with its default hyperparameter. We used batch size 32 for each worker and used 16 workers.

Table 2 summarizes the results. In this example as well as the previous CIFAR10 example, Variance-based Gradient Compression shows a significantly high compression ratio, with comparable accuracy. While in this case, Strom's method's accuracy was comparable with no compression, given

the significant accuracy degradation with Strom's method on CIFAR10, it appears Variance-based Gradient Compression provides a more robust solution. Note that the training configuration with MomentumSGD is highly optimized to training without any compression. For reference, the original paper of ResNet-50 reports its accuracy as 75.3% (He et al., 2016). Wen et al. (2017) reports that it caused up to 2% accuracy degradation in training with GoogLeNet (Szegedy et al., 2015) on ImageNet and our method causes no more degradation compared to quantization-based approaches.

## 7 CONCLUSION

We proposed a novel method for gradient compression. Our method can reduce communication cost significantly with no or only slight accuracy degradation. Contributions of our work can be summarized in the following three points. First, we proposed a novel measurement of ambiguity (high variance, low amplitude) to determine when a gradient update is required. Second, we showed the application of this measurement as a threshold for updates significantly reduces update requirements, while providing comparable accuracy. Third, we demonstrated this method can be combined with other efficient gradient compression approaches to further reduce communication cost.

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

## A A SIMPLIFIED VIEW OF THE VARIANCE-BASED CRITERION

We derive that the criterion (3) corresponds to the criterion (1).

$$\left(\sum_{z \in B} \frac{1}{|B|} \nabla_i f_z(x)\right)^2 > \alpha \sum_{z \in B} \left(\frac{1}{|B|} \nabla_i f_z(x)\right)^2$$

$$\iff \quad \frac{|B| - \alpha}{|B|} \left(\sum_{z \in B} \frac{1}{|B|} \nabla_i f_z(x)\right)^2 > \alpha \frac{1}{|B|} \left(\frac{1}{|B|} \sum_{z \in B} (\nabla_i f_z(x)) - \sum_{z \in B} \left(\frac{1}{|B|} \nabla_i f_z(x)\right)^2\right)$$

$$\iff \quad \frac{|B| - \alpha}{|B|} \left(\sum_{z \in B} \frac{1}{|B|} \nabla_i f_z(x)\right)^2 > \alpha \frac{1}{|B|^2} \sum_{z \in B} \left(\nabla_i f_z(x) - \frac{1}{|B|} \sum_{z \in B} (\nabla_i f_z(x))\right)^2$$

$$\iff \quad \frac{|B| - \alpha}{|B|} \left(\sum_{z \in B} \frac{1}{|B|} \nabla_i f_z(x)\right)^2 > \alpha \frac{|B| - 1}{|B|^2} \frac{1}{|B| - 1} \sum_{z \in B} \left(\nabla_i f_z(x) - \frac{1}{|B|} \sum_{z \in B} (\nabla_i f_z(x))\right)^2$$

$$\iff \quad \frac{|B| - \alpha}{|B|} \left(\sum_{z \in B} \frac{1}{|B|} \nabla_i f_z(x)\right)^2 > \alpha \frac{|B| - 1}{|B|^2} V_B[\nabla_i f_z(x)]$$

$$\iff \quad \nabla_i f_B(x)^2 > \alpha \frac{|B| - 1}{|B| - \alpha} \frac{1}{|B|} V_B[\nabla_i f_z(x)]$$

$V_B[\nabla_i f_z(x)]/|B|$ is an estimated variance of the means of gradients in mini-batches with size $|B|$. For $\alpha = 1$, the above criterion reduces to

$$\nabla f_B(x)^2 > \frac{1}{|B|} V_B[\nabla f_z(x)],$$

which is an estimated version of the sufficient condition (2). The term $(|B| - 1)/(|B| - \alpha)$ is approximately 1 in most settings.

## B RUNNING EXAMPLE OF QUANTIZATION

Let $(0.04, 0.31, -6.25, 22.25, -35.75)$ be a part of gradient elements corresponding to a matrix. Sign bits are separately processed and we consider their absolute values here: $(0.04, 0.31, 6.25, 22.25, 35.75)$. Now, $M_k$ is the max of the elements: 35.75. $2^{\lfloor \log_2 M_k \rfloor}$ is 32. After rounding, $g_{i'}$ of each element become $0.03125, 0.25, 8, 16, 32$ and $d_i$ for each element become $10, 7, 2, 1, 0$. Note that higher $d_i$ corresponds to smaller $g'_i$. We can use only 3 bits and thus

we cannot represent 10 and it will not be sent, which means it will not be sent. Finally, we send them with $\lfloor \log_2 M_k \rfloor$, sign bits and index: $\{\lfloor \log_2 M_k \rfloor : 5, ((+7, \text{index} : 1), (-2, \text{index} : 2), (+1, \text{index} : 2), (-0, \text{index} : 3))\}$.

## C VISUAL REPRESENTATION OF EXPERIMENTAL RESULTS

Figure 3 are scatter plots of Table 1 and 2. The upper right corner is desirable. The figures suggests superiority of our variance-based compression and hybrid algorithm.

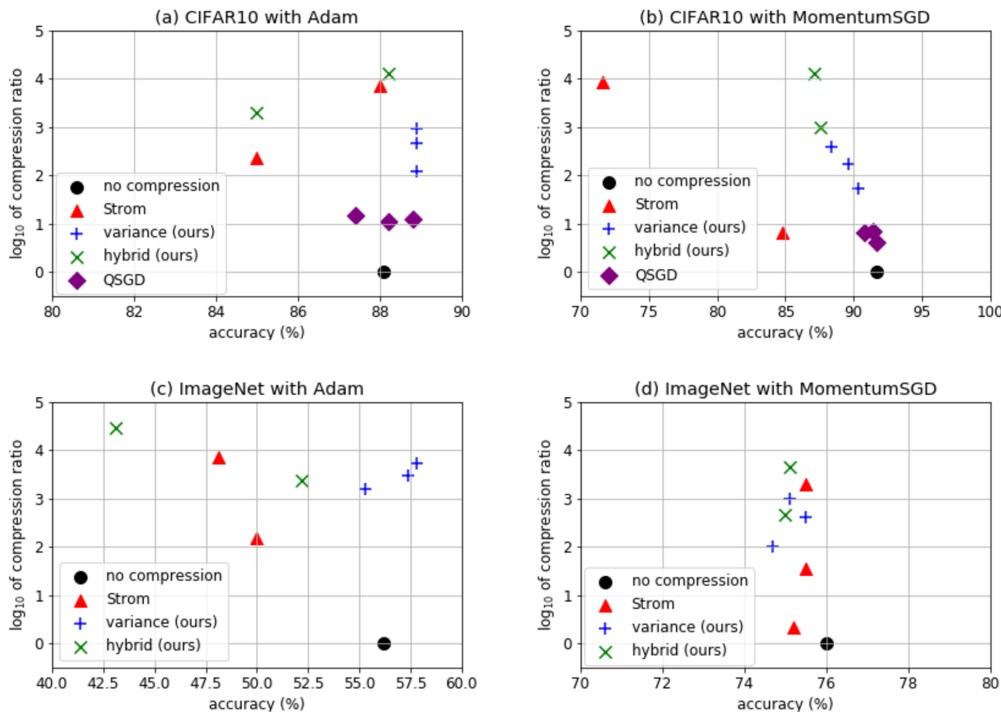

Figure 3: Scatter plots of relation between accuracy and compression ratio. The four plots correspond to the configurations on datasets and optimization methods as follows: (a) CIFAR-10 and Adam, (b) CIFAR-10 and MomentumSGD, (c) ImageNet and Adam, (d) ImageNet and MomentumSGD. No compression denotes the case where we do not use any compression methods. Variance is a method described in Sec. 4.1 and 4.2. Hybrid is a combination of variance based compression and Strom's algorithm. Some outliers are not plotted. Upper right of these figures is desirable for all compression algorithms.

## D ARCHITECTURE OF VGG-LIKE NETWORK USED ON CIFAR-10

Table 3 shows the network architecture used for experiments on CIFAR-10. All convolutional layers are followed by batch normalization and ReLU activation. The code is available in examples of Chainer (Tokui et al., 2015) on GitHub.

Table 3: The network architecture for the CIFAR-10 classification task

| input (3x32x32) |
| :---: |
| conv3-64
dropout(0.3)
conv3-64 |
| maxpool |
| conv3-128
dropout(0.4)
conv3-128 |
| maxpool |
| conv3-256
dropout(0.4)
conv3-256
dropout(0.4)
conv3-256 |
| maxpool |
| conv3-512
dropout(0.4)
conv3-512
dropout(0.4)
conv3-512 |
| maxpool |
| conv3-512
dropout(0.4)
conv3-512
dropout(0.4)
conv3-512 |
| maxpool |
| dropout(0.5)
fully connected 512
bn
relu |
| dropout(0.5)
fully connected 10 |
| softmax |

