# OpenReview forum: "Variance-based Gradient Compression for Efficient Distributed Deep Learning"
_ICLR.cc/2018/Conference — Invite to Workshop Track_

### Official Review · AnonReviewer3 · 2017-11-27
**The idea to adopt approximated variances of gradients to reduce communication cost seems to be interesting. However, there also exist several major issues in the paper.**

**Rating:** 6
**Confidence:** 4

**Review:**

This paper proposes a variance-based gradient compression method to reduce the communication overhead of distributed deep learning. Experiments on real datasets are used for evaluation.

The idea to adopt approximated variances of gradients to reduce communication cost seems to be interesting. However, there also exist several major issues in the paper.

Firstly, the authors propose to combine two components to reduce communication cost, one being variance-based gradient compression and the other being quantization and parameter encoding. But the contributions of these two components are not separately analyzed or empirically verified.

Secondly, the experimental results are unconvincing. The accuracy of Momentum SGD for ‘Strom, \tau=0.01’ on CIFAR-10 is only 10.6%. Obviously, the learning procedure is not convergent. It is highly possible that the authors do not choose a good hyper-parameter. Furthermore, the proposed method (not the hybrid) is not necessarily better than Strom except for the case of Momentum SGD on CIFAR-10. Please note that the case of Momentum SGD on CIFAR-10 may have a problematic experimental setting for Strom. In addition, it is weird that the experiment on ImageNet does not adopt the same setting as that on CIFAR-10 to evaluate both Adam and Momentum SGD.

---

> ### Author Response · Authors · 2017-12-25
> **Thank you for your review.**
>
> Thank you for your review. We are glad to hear that you found our algorithm interesting.
>
> > But the contributions of these two components are not separately analyzed or empirically verified.
> Thank you for your comment. The main contribution is intended to be the variance-based gradient compression, with the quantization provided as a way to fit both values of gradient elements and its index in 32-bit while not rounding many elements to zero. We amended our paper with the following:
> Sec 4.2’’’To allow for comparison with other compression methods, we propose a basic quantization process. …’’’
>
> > The accuracy of Momentum SGD for ‘Strom, \tau=0.01’ on CIFAR-10 is only 10.6%. Obviously, the learning procedure is not convergent. It is highly possible that the authors do not choose a good hyper-parameter.
> Thank you for your comment. We amended our paper with the following:
> Sec.6.1’’’We note that we observed unstable behaviors with other thresholds around 0.01.’’’
> Appendix D’’’The code is available in examples of Chainer on GitHub.’’’
>
> > Furthermore, the proposed method (not the hybrid) is not necessarily better than Strom except for the case of Momentum SGD on CIFAR-10. Please note that the case of Momentum SGD on CIFAR-10 may have a problematic experimental setting for Strom.
> Thank you for your comment. We amended our paper with the following:
> Sec. 6.1’’’We also would like to mention the difficulty of hyperparameter tuning in Strom's method. … On the other hand, our algorithm is free from such problem. Moreover, when we know good threshold for Strom's algorithm, we can just combine ours to get further compression.’’’
>
> > In addition, it is weird that the experiment on ImageNet does not adopt the same setting as that on CIFAR-10 to evaluate both Adam and Momentum SGD.
> Thank you for your comment. We amended our paper with the following:
> Sec 6.2 ‘’’We also evaluated algorithms with replacing MomentumSGD and its learning rate scheduling to Adam with its default hyperparameter.’’’

---

### Official Review · AnonReviewer2 · 2017-11-27
**Ok but not good enough**

**Rating:** 4
**Confidence:** 4

**Review:**

The paper proposes a novel way of compressing gradient updates for distributed SGD, in order to speed up overall execution. While the technique is novel as far as I know (eq. (1) in particular), many details in the paper are poorly explained (I am unable to understand) and experimental results do not demonstrate that the problem targeted is actually alleviated.

More detailed remarks:
1: Motivating with ImageNet taking over a week to train seems misplaced when we have papers claiming to train ImageNet in 1 hour, 24 mins, 15 mins...
4.1: Lemma 4.1 seems like you want B > 1, or clarify definition of V_B.
4.2: This section is not fully comprehensible to me.
- It seems you are confusingly overloading the term gradient and words derived (also in other parts or the paper). What is "maximum value of gradients in a matrix"? Make sure to use something else, when talking about individual elements of a vector (which is constructed as an average of gradients), etc.
- Rounding: do you use deterministic or random rounding? Do you then again store the inaccuracy?
- I don't understand definition of d. It seems you subtract logarithm of a gradient from a scalar.
- In total, I really don't know what is the object that actually gets communicated, and consequently when you remark that this can be combined with QSGD and the more below it, I don't understand it. This section has to be thoroughly explained, perhaps with some illustrative examples.
4.3: allgatherv remark: does that mean that this approach would not scale well to higher number of workers?
4.4: Remarks about quantization and mantissa manipulation are not clear to me again, or what is the point in doing so. Possible because the problems above.
5: I think this section is not too useful unless you can accompany it with actual efficient implementation and contrast the practical performance.
6: Given that I don't understand how you compress the information being communicated, it is hard to believe the utility of the method. The objective was to speed up training time because communication is bottleneck. If you provide 12,000x compression, is it any more practically useful than providing 120x compression? What would be the difference in runtime? Such questions are never discussed. Further, if in the implementation you discuss masking mantissa, I have serious concern about whether the compression protocol is feasible to implement efficiently, without writing some extremely low-level code. I think the soundness of work addressing this particular problem is damaged if not implemented properly (compared to other kinds of works in current ML related research). Therefore I highly recommend including proper time comparison with a baseline in the future.
Further, I don't understand 2 things about the Tables. a) how do you combine the proposed method with Momentum in SGD? This is not discussed as far as I can see. b) What is "QSGD, 2bit" If I remember QSGD protocol correctly, there's no natural mapping of 2bit to its parameters.

---

> ### Author Response · Authors · 2017-12-25
> **Thank you for your review.**
>
> Thanks for the review. We're glad to hear that you found our technique to be novel. We've amended our paper in light of your review. We hope this helps explain the details and demonstrate how our technique alleviates the problem of transmitting gradients between nodes.
>
> Section 1
> > Motivating with ImageNet taking over a week to train seems misplaced when we have papers claiming to train ImageNet in 1 hour, 24 mins, 15 mins...
> Thanks for the comment. We have amended our paper with the following:
> ‘’’For example, it takes over a week to train ResNet-50 on the ImageNet dataset if using a single GPU. … For example, when using 1000BASE-T Ethernet, communication takes at least ten times longer than forward and backward computation for ResNet-50, making multiple nodes impractical. High performance interconnections such as InfiniBand and Omni-Path are an order of magnitude more expensive than commodity interconnections, which limits research and development of deep learning using large-scale datasets to a small number of researchers.’’’
>
> > 4.1: Lemma 4.1 seems like you want B > 1, or clarify definition of V_B.
> Correct. We have amended our paper with the following:
> ‘’’Lemma 4.1
>  A sufficient condition that a vector -g is a descent direction is
>       \|g - \nabla f(x)\|_2^2 < \|g\|_2^2.
> We are interested in the case of g = \nabla f_B(x), the gradient vector of the loss function over B.
> By the weak law of large numbers, when B > 1, the left-hand side with g = \nabla f_B(x) can be estimated as follows.’’’
> Note, \nabla_B f(x) in lemma 4.1 of our first paper was replaced with a symbol g.
>
> > 4.2: This section is not fully comprehensible to me.
> > - It seems you are confusingly overloading the term gradient and words derived (also in other parts or the paper). What is "maximum value of gradients in a matrix"? Make sure to use something else, when talking about individual elements of a vector (which is constructed as an average of gradients), etc.
> You’re right. We amended the paper to replace gradient with ‘gradient element’ when we refer elements of gradient vectors.
> ‘’’Our quantization except for the sign bit is as follows. For a weight matrix W_k (or a weight tensor in CNN), there is a group of gradient elements corresponding to the matrix. Let M_k be the maximum absolute value in the group.’’’
>
> > - Rounding: do you use deterministic or random rounding? Do you then again store the inaccuracy?
> Good questions. We amended our paper as follows:
> Sec 4.2’’’... We do not adopt stochastic rounding like QSGD nor accumulate rounding error g_i - g'_i for the next batch because this simple rounding does not harm accuracy empirically.’’’
>
> > - I don't understand definition of d. It seems you subtract logarithm of a gradient from a scalar.
> d is a difference of two scalars.
> >  In total, I really don't know what is the object that actually gets communicated, and consequently when you remark that this can be combined with QSGD and the more below it, I don't understand it. This section has to be thoroughly explained, perhaps with some illustrative examples.
> We hope our clarification between ‘gradient’ and ‘gradient element’ made the definition of d clearer. We amended our paper as follows:
> Sec 4.2‘’’... After deciding which gradient elements to send, each worker sends pairs of a value of a gradient element and its parameter index …’’’
> Sec 4.2’’’... Because the variance-based sparsification method described in subsection 4.1 is orthogonal to the quantization shown above, we can reduce communication cost further using sparsity promoting quantization methods such as QSGD instead.’’’
>
> > 4.3: allgatherv remark: does that mean that this approach would not scale well to higher number of workers?
> It does scale well. We amended our paper with the following:
> Sec 4.3’’’Thanks to the high compression ratio possible with this algorithm in combination with other compression methods, even large numbers of workers can be supported.’’’
>
> > 4.4: Remarks about quantization and mantissa manipulation are not clear to me again, or what is the point in doing so. Possible because the problems above.
> To make a point of the mantissa operations clear, we amended our paper with the following:
> ‘’’The quantization of parameters described in subsection 4.2 can also be efficiently implemented with the standard binary floating point representation using only binary operations and integer arithmetic as follows.’’’

---

> > ### Author Response · Authors · 2017-12-25
> > **Thank you for your review. (2)**
> >
> > > 5: I think this section is not too useful unless you can accompany it with actual efficient implementation and contrast the practical performance.
> > Yes, we would like to be able to do this comparison. We amended the paper to include this at the beginning of Section 5 -- Performance Analysis:
> > ‘’’Because common deep learning libraries do not currently support access to gradients of each sample, it is difficult to contrast practical performance of an efficient implementation in the commonly used software environment.In light of this, we estimate speedup of each iteration by gradient compression with a performance model of communication and computation.’’’
> >
> > > If you provide 12,000x compression, is it any more practically useful than providing 120x compression?
> > Yes, we believe it can be practically useful, depending on the underlying computation infrastructure. With existing compression methods, computation with a large number of nodes essentially requires high bandwidth connections like InfiniBand. Much higher levels of compression make it possible to consider large numbers of nodes even with commodity-level bandwidth connections. We amended our paper with the following:
> > Sec 6.1’’’The hybrid algorithm's compression ratio is several orders higher than existing compression methods with a low reduction in accuracy. This indicates the algorithm can make computation with a large number of nodes feasible on commodity level infrastructure that would have previously required high-end interconnections.’’’
> > Sec 6.2’’’In this example as well as the previous CIFAR10 example, Variance-based Gradient  Compression shows a significantly higher compression ratio, with comparable accuracy. While in this case, Strom's method's accuracy was comparable with no compression, given the significant accuracy degradation with Strom's method on CIFAR10, it appears Variance-based Gradient Compression provides a more robust solution.’’’
> >
> > > Further, if in the implementation you discuss masking mantissa, I have serious concern about whether the compression protocol is feasible to implement efficiently, without writing some extremely low-level code.
> > Yes, low-level code would be required to use our method. It is also true for other existing methods.
> >
> > > Therefore I highly recommend including proper time comparison with a baseline in the future. Once parameter variance is provided within one of the standard calculation libraries of primitives for neural deep neural networks, this time comparison can be done.
> >
> > > a) how do you combine the proposed method with Momentum in SGD? This is not discussed as far as I can see.
> > We amended our paper with the following:
> > Sec. 4.1 ‘’’... In the combination with optimization methods like Momentum SGD, gradient elements not sent are assumed to be equal to zero.’’’
> >
> > > b) What is "QSGD, 2bit" If I remember QSGD protocol correctly, there's no natural mapping of 2bit to its parameters.
> > Thank you for your comment. We misunderstood meaning ‘bit” used in experiment section of original QSGD paper. We asked the authors at NIPS, and we reran experiments. We amended our paper as follows:
> > Sec 6.1’’’We used two's complement in implementation of QSGD and "bit" represents the number of bits used to represent each element of gradients. "d" represents a bucket size.’’’

---

> > > ### Comment · AnonReviewer2 · 2018-01-02
> > > **Thanks for update**
> > >
> > > After seeing your response, and reviews of other reviewers, my opinion is still that this is an interesting work, but more needs to be done to publish it.
> > >
> > > In particular, you propose something that you show is an interesting thing to do, but you do not demonstrate that this is actually a useful thing to do. This is very important difference for the specific problem you try to address. Comments such as "yes, we believe it can be practically useful" are in my opinion deeply insufficient, and the belief should be explicitly captured in experimental results. This is what I would suggest to focus on in a revision.

---

> > > > ### Author Response · Authors · 2018-01-05
> > > > **Thank you for your response.**
> > > >
> > > > First of all, thank you for reading our response and giving us an additional comment. To support our arguments, we estimate actual speedup by variance-based compression using micro-benchmarks of communication over slow interconnection.
> > > > First, we measured computation and communication time for training ResNet50 on 16 nodes using Infiniband. It took 302.72 ms for computation and 69.95 ms for communication for each iteration in average. We note that ResNet50 contains about 102 MB of parameters.
> > > > Next, we measured communication time of allreduce without compression and that of allgatherv with compression. We used 16 t2.micro instances of AWS Its point-to-point bandwidth was about 100MB/s. We used OSU micro-benchmarks (http://mvapich.cse.ohio-state.edu/benchmarks/) for the measurements. Summary of the result is the following:
> > > > --
> > > > allreduce
> > > > compression | Avg elapsed time (ms)
> > > > 1 | 9,572.95
> > > > --
> > > > allgatherv
> > > > compression | Avg elapsed time (ms)
> > > > 10 | 3,440.70
> > > > 100 | 314.17
> > > > 1,000 | 30.09
> > > > 10,000 | 4.26
> > > > --
> > > > With this result, we can see that communication takes longer time compared to usual computation time even with 100x compression. Thus, we can say that even with only 16 nodes, compression ratio over a hundred is desirable to achieve high scalability. In use cases with more nodes, communication will take longer and thousands of times of compression will help. We hope this addresses your concern.

---

### Official Review · AnonReviewer1 · 2017-11-27
**Simple yet efficient new algorithm for gradient compression with good performance.**

**Rating:** 7
**Confidence:** 4

**Review:**

The authors propose a new gradient compression method for efficient distributed training of neural networks. The authors propose a novel way of measuring ambiguity based on the variance of the gradients. In the experiment, the proposed method shows no or slight degradation of accuracy with big savings in communication cost. The proposed method can easily be combined with other existing method, i.e., Storm (2015), based on the absolute value of the gradient and shows further efficiency.

The paper is well written: clear and easy to understand. The proposed method is simple yet powerful. Particularly, I found it interesting to re-evaluate the variance with (virtually) increasing larger batch size. The performance shown in the experiments is also impressive.

I found it would have also been interesting and helpful to define and show a new metric that incorporates both accuracy and compression rate into a single metric, e.g., how much accuracy is lost (or gained) per compression rate relatively to the baseline of no compression. With this metric, the comparison would be easier and more intuitive.

---

> ### Author Response · Authors · 2017-12-25
> **Thank you for your review.**
>
> Thank you for your review and helpful suggestion.
> We tried to make a new single metric, however, we are not sure how to combine accuracy and compression ratio as they are not directly comparable.
> To make a comparison between methods more intuitive, we added scatter plots of accuracy and compression ratio in Appendix C.

---

### Author Response · Authors · 2017-12-25
**Eratta**

We found that we used mistakenly smaller \zeta for our algorithm than the value specified in our paper, and thus we reran experiments and updated experimental results.
We also found inconsistency of our setting for QSGD with its original paper, and we corrected the experimental results.

---

### Public Comment · ~Felix_Yu1 · 2018-01-29
**An earlier work on reducing communication cost in distributed SGD**

Dear authors,

I am writing to bring your attention to the following work
Suresh, Yu, Kumar, McMahan Distributed mean estimation with limited communication ICML 2017

In this paper, we showed a few variants to reduce the communication cost of distributed mean estimation. One method is communication optimal. The methods and analysis are directly applicable to distributed SGD, as each step of SGD involves distributed mean estimation of gradients.

Link to our theory paper (distributed mean estimation): (http://proceedings.mlr.press/v70/suresh17a/suresh17a.pdf) and link to our implementation workshop paper (https://arxiv.org/abs/1610.05492).

---

### Decision · Program_Chairs · 2018-01-29
**ICLR 2018 Conference Acceptance Decision**

**Decision:**

Invite to Workshop Track

**Comment:**

The reviewers find the gradient compression approach novel and interesting, but they find the empirical evaluation not fully satisfactory. Some aspects of the paper have improved with the feedback from the reviewers, but because of the domain of the paper, experimental evaluation is very important. I recommend improving the experiments by incorporating the reviewers' comments.